# Nanocarriers as a Delivery Platform for Anticancer Treatment: Biological Limits and Perspectives in B-Cell Malignancies

**DOI:** 10.3390/pharmaceutics14091965

**Published:** 2022-09-17

**Authors:** Sara Bozzer, Michele Dal Bo, Maria Cristina Grimaldi, Giuseppe Toffoli, Paolo Macor

**Affiliations:** 1Department of Life Sciences, University of Trieste, 34127 Trieste, Italy; 2Experimental and Clinical Pharmacology Unit, Centro di Riferimento Oncologico di Aviano (CRO), Istituto di Ricovero e Cura a Carattere Scientifico (IRCCS), 33081 Aviano, Italy

**Keywords:** nanocarriers, delivery, cancer, polymeric NPs

## Abstract

Nanoparticle-based therapies have been proposed in oncology research using various delivery methods to increase selectivity toward tumor tissues. Enhanced drug delivery through nanoparticle-based therapies could improve anti-tumor efficacy and also prevent drug resistance. However, there are still problems to overcome, such as the main biological interactions of nanocarriers. Among the various nanostructures for drug delivery, drug delivery based on polymeric nanoparticles has numerous advantages for controlling the release of biological factors, such as the ability to add a selective targeting mechanism, controlled release, protection of administered drugs, and prolonging the circulation time in the body. In addition, the functionalization of nanoparticles helps to achieve the best possible outcome. One of the most promising applications for nanoparticle-based drug delivery is in the field of onco-hematology, where there are many already approved targeted therapies, such as immunotherapies with monoclonal antibodies targeting specific tumor-associated antigens; however, several patients have experienced relapsed or refractory disease. This review describes the major nanocarriers proposed as new treatments for hematologic cancer, describing the main biological interactions of these nanocarriers and the related limitations of their use as drug delivery strategies.

## 1. Introduction

The eradication of diseases remains an elusive clinical goal, mainly to the heterogeneous and idiosyncratic nature of the individual disease and the inability to target therapeutics to pathological areas without damaging normal tissues. Therefore, the ideal drug is easy to administer, binds univocally to its specific target with high specificity and affinity, does not modulate off-target functions, and persists in the body only for the time necessary to have a therapeutic effect. The complex nature of cancer presents multiple challenges to its treatment. The gold standard of cancer treatment is still represented by chemotherapeutic drugs; however, this therapeutic strategy still presents inherent challenges. Among them, the most important is depicted by the poor accumulation in the diseased microenvironment, due to the lack of specificity. Hence, undesired side effects in healthy tissues occur, especially in the heart, bone marrow, gastrointestinal tract, and nervous system [1,2].

Over the last years, nanomedicine has become an inescapable part of modern everyday life, and most of the limits of traditional drugs can be bypassed by nanomedicine, which uniquely focuses on medically related, patient-centric nanotechnologies [3] (Figure 1).

The European Science Foundation (ESF) has defined nanomedicine as a tool that uses a nano-sized device for the diagnosis, prevention, and treatment of diseases. Therefore, the focus of nanomedicine is to improve the quality of life of patients, encompassing the three main nanotechnology areas, which are developed for healthcare applications such as diagnostics, imaging agents, and drug delivery through innovative technologies and biomaterials. In this sense, the European Commission (EC) published the ‘Recommendation on the definition of nanomaterials’ that defines whether a material can be considered a nanomaterial or not in terms of legislation and policy in the European Union [4]. According to the EC recommendation, “Nanomaterial” represents a material containing particles, aggregate or not and, where, for at least 50% of the particles, one or more external dimensions is in the size range 1–100 nm. In this context, nanoparticles (NPs) for medical applications are defined as particles with a size between 1 and 1000 nm [5,6,7], and they can be engineered to have different sizes, shapes, chemical compositions, surface chemical characteristics, and hollow or solid structures leading to them being suitable for the treatment of several diseases [8]. Moreover, nanomaterials, differently from other widespread materials, have a high ratio of surface area to volume, as well as electronic, magnetic, and biological properties. Indeed, all these features allow the development of personalized and safe drugs. Considerable technological successes have been achieved in this field, but the main obstacles to the widespread application of nanomedicine derive from the intrinsic features of the tumor and an incomplete understanding of nano–bio interactions [9]. With the progression of nanodevices, the idea is to include the optimal dose of a drug in a nano-system that actively reaches its target and here releases its content, improving efficacy. At the same time, the specificity for tumor cells prevents the interactions of NPs with healthy tissues, thus reducing side effects. In addition, these systems are able to protect themselves from degradation in vivo, prolonging their availability [1]. Thus, the choice of an appropriate nanocarrier is based on the more suitable application.

A major public health problem worldwide is represented by cancer, and it is still the second leading cause of death in the United States, even if in 2020 the diagnosis and treatment of cancer were hampered by the coronavirus disease 2019 (COVID-19) pandemic [10]. Childhood and juvenile cancer, which affects individuals between 0 and 19 years old, consists of a set of diseases that have their characteristics concerning to histological type and clinical behavior of the disease. According to the World Health Organization (WHO), “Cancer is a leading cause of death for children and adolescents” (World Health Organization 2021). In particular, in the States, cancer is the second most common cause of death among children, surpassed only by accidents [10]. Moreover, in most populations, childhood and juvenile cancer accounts for up to 4% of all malignant tumors; however, in developing countries, where the child population represents a bigger portion of the population, cancer accounts for up to 10% of all cancers [11]. Among them, leukemia is the most common childhood cancer, accounting for 28% of cases, followed by brain and other nervous system tumors (27%) [10]. Globally, acute lymphoblastic leukemia (ALL) is the most common childhood cancer and is estimated to account for 19% of total childhood cancer incidence, followed by non-Hodgkin lymphoma (NHL), Burkitt lymphoma (BL), and retinoblastoma (Figure 2).

Moreover, summing productive years lost due to the mortality or disability of the pathology, over a third of them are caused by leukemia [12]. As predictable, the most common cancer differs by region; for example, the incidence of ALL is significantly lower in sub-Saharan Africa than in other regions in contrast to statistics of BL, which are inverted if compared. Since it is generally not possible to prevent malignancies in children, the most effective strategy to reduce the burden of cancer and improve outcomes is to focus on a prompt, correct diagnosis followed by the most effective treatment available [13].

This review describes nanocarriers that have been proposed mainly as carriers for cancer treatments. Particular attention is paid to their composition and structure. In addition, the main biological interactions of nanocarriers and the related limitations of their use as drug delivery strategies are described. Since a relevant number of targeted therapies, such as immunotherapies using monoclonal antibodies, have been approved for the treatment of hematologic neoplasms, a description of several tumor-associated antigens as potential targets for drug delivery strategies for the treatment of hematologic neoplasms is proposed.

## 2. Nanocarriers as Delivery Platforms

Delivering drugs loaded inside NPs to cancer cells is a highly complicated process. Perhaps the right attitude could be to embrace the benefits of nanotechnologies but acknowledge their limitations [1]; indeed, it is necessary that NPs evade immune surveillance by avoiding opsonization by serum proteins and selectively extravasate at the tumor site [14,15,16]. Among the essential NPs’ features to achieve homogeneous distribution throughout a tumor, there is the ability to overcome thick tumor stroma, uptake by macrophages, high interstitial fluid pressure, and slow diffusion [17]. Among different nanostructure morphologies for drug delivery, NP-based drug delivery shows numerous advantages for controlling the release of biological factors. These advantages include the possibility of adding a selective targeting mechanism, the controlled release, protection of deliverable agents, the extension of the circulation time in the body, and the functionalization of the NPs, which help to achieve the best possible result. Luckily, in the research landscape, there are plenty of different nano-delivery systems which increase the possibility of better investigating cancer and choosing the more suitable vector for therapy. A vast choice of materials, differing from natural to synthetic and to hybrid polymers, has been developed in the form of delivery vesicles for anticancer treatment. Various biomaterials were investigated for constructing therapeutic delivery carriers through the customization of their chemical and physical properties to meet specific needs in different clinical applications. Nanocarriers, owing to their high surface-area-to-volume ratio, make it possible to achieve high ligand density on the surface for targeting purposes [18]. Furthermore, changing the basic properties of NPs (i.e., diameter, shape, and surface charge) enable modulating immunotherapy. Indeed, NPs with a diameter of over 500 nm can target macrophages and be internalized; on the other hand, the smaller NPs (<50 nm) have an enhanced ability to elicit the immune activities over larger NPs (>100 nm), because the smaller ones tend to traffic to lymph nodes, whereas the larger ones are hindered [19]. These systems can also be used to increase local drug concentration by carrying the drug within tumor cells and inducing the controlled release of the payload after binding to and internalizing it into the target cells [20]. NP-based drug delivery systems for systemic (i.e., intravenous) applications have significant advantages over their free drug counterparts, showing nanomedicine and nanotechnologies not as a futuristic revolution but as something already in action. In the last twenty-five years, the FDA has approved different types of NPs for cancer treatment, and, at present, many nanostructures are under investigation [1,2,21,22].

Since the early 1970s, when Gregoriadis et al. established the concept that liposomes could be used as drug carriers, just like “putting old drugs into new clothing” [23], several different kinds of nanocarriers were produced, characterized, and finally used for many different purposes: liposomes, micelles, dendrimers, carbon nanomaterials, inorganic NPs, and polymeric NPs (Figure 3) [19].

### 2.1. Composition of Nano-Systems

Liposomes (Figure 3a) are artificial vesicles composed of a uni- or multi-layer(s) of phospholipids that can incorporate hydrophilic compounds in their aqueous compartment(s) and, possibly, hydrophobic compounds in their lipid bilayer(s). Liposomes are biodegradable and easily modifiable both with targeting agents to improve the selectivity and with coating agents to be made more biocompatible. Moreover, their surface can be modified to prolong their circulation and prevent their elimination from the circulatory system with the aid of natural or biocompatible polymers to shield them from opsonins, which are proteins able to bind foreign substances or cells, inducing their phagocytosis. The liposome dimension is between 30 nm and 50 μm. They are spherical vesicles, able to self-assemble in an aqueous solution due to their amphiphilic nature. The polar shell is capable of holding lipophilic molecules, and, by contrast, the aqueous core can be loaded with hydrophilic molecules [24]. Just like other nanostructures, their surface can be functionalized with different molecules in order to target specific cells or tissues. Despite the success of liposome-based nanocarriers as drug delivery systems, there are some limitations in their widespread application. Indeed, liposomes are characterized by low stability in long-term storage, and the amount of loadable materials is limited [25].

Liposomes can release their content in response to a specific trigger signal (such as hyperthermia, pH variations, an alternation of the external magnetic field, or ultrasound). This triggered release is extremely useful to avoid side effects and increment therapeutic efficacy. Since the first description, they have experienced exponential growth [23,26]. Particularly, in 1995 a formulation based on these nano-vectors loaded with doxorubicin and known as Doxil^®^ was approved by the FDA [27]. This formulation demonstrated an improved circulation half-life of doxorubicin and an increased accumulation in the tumor tissue [28]. To this day, more than eighteen liposomal drugs have been approved by the FDA for the treatment of cancer and many other diseases [24].

Polymeric micelles (Figure 3b) are self-assembled nanostructures made by amphiphilic block copolymers, which aggregate in order to reduce contact with water molecules producing vesicles [24]. These structures are usually found in applications regarding the delivery of hydrophobic drugs, which can be loaded into their core. Micelles can also be used to carry small organic molecules, peptides, carbohydrates, monoclonal antibodies (mAbs), and DNA/RNA aptamers. Even if not yet widely utilized clinically, micelles have several advantages, such as being a facility to be prepared and loaded with drug agents, biocompatibility, and high, although limited, stability in biological fluids [29].

Differently, dendrimers (Figure 3c) represent a relatively new field in polymer chemistry. They are synthetic, tree-like structures fabricated through a stepwise process, which yields molecules possessing a core, interior layers, and an exterior with terminal functionalities attached to the outmost generations. The outer can be easily conjugated with targeting molecules, imaging agents, and drugs due to the high water solubility and well-defined chemical structures. These features, together with their biocompatibility, render dendrimers promising for biomedical applications. Since 1984, when the first article introduced the term “dendrimer”, a variety of dendrimers have been investigated for cancer treatment, but these are more expensive than other NPs and require many repetitive steps for synthesis, posing a challenge for large-scale production [20,27,30].

Carbon nanotubes (Figure 3d) have also been analyzed for cancer treatment. These structures can be linked to various biological materials and enter cells via endocytosis. Single-walled carbon nanotubes (SWCNTs) are suitable for biological applications thanks to the high stability of the suspensions in physiological buffers. Treatments based on the use of SWCNTs are promising due to their interesting optical properties; in fact, SWCNTs can be used for photothermal and photodynamic therapy [27,31]. Carbon nanotubes were demonstrated to induce severe side effects such as asbestos-like inflammation and granulomas in female mice; moreover, severe impacts on the cardiovascular system were reported with specific types of carbon-based nanomaterials, such as single- or multi-walled carbon nanotubes [30].

NPs can be divided into two classes: organic and inorganic NPs. Inorganics NPs (Figure 3e) are made of metal atoms (metallic) or a mixture of metal and non-metal atoms (semi-conducting). Each of these types of NPs show peculiar features and are used for different applications. Metallic NPs usually find medical applications as diagnostic probes for positron emission tomography (PET), magnetic resonance imaging (MRI), optical imaging, and X-ray. The strength and durability of these materials are unique properties that make them suitable as delivery tools [32]. Inorganic crystalline NPs approved by the FDA are limited to hydroxyapatite and calcium phosphate for use as bone graft substitutes; nevertheless, nanocrystals are helpful in overcoming the solubility issues related to several drug compounds and are marketed for a range of indications [32].

Another type of nanocarrier, frequently considered as an alternative to liposomal vehicles, is represented by organic NPs; they can be produced using different polymers or lipids, resulting in biodegradable and efficient drug delivery systems [22]. Polymeric NPs (Figure 3f) are attractive because of their high in vivo stability and loading efficiency; as a consequence, polymers result as the most common materials for constructing NP-based drug carriers. One of the earliest reports of their use for cancer therapy dates to 1979. Polymeric NPs are structurally defined by a core and a shell. The therapeutic agent could be conjugated to the surface of the NP or encapsulated and protected inside the polymeric core [33]. These nanostructures can be assembled using both synthetic and natural polymers, allowing the generation of a variety of applications such as imaging, detection of apoptosis, and drug delivery [34,35].

#### 2.1.1. Biodegradable Polymers

Since 1976, when Langer and Folkman demonstrated that biodegradable polymers can be used for controlled release, this kind of nanodevice earned considerable success [36]. Despite the progress in the development of lipid-based NPs, polymeric materials are increasingly emerging as a better system of controlled and prolonged drug release. The process and its extent depend upon parameters relating to the nature of the carrier system. For example, the payload can diffuse through the polymer wall with a diffusion rate that is dependent on the degree of crosslinking of the matrix. Alternatively, the release can be caused by the erosion of the NPs’ surface or polymer matrix degradation. However, both processes depend on a progressive alteration of the NPs’ structure, as the degradation of the entire polymeric network results in greater drug release [36]. Following this discovery, considerable successes have been achieved in the therapeutic field, and, over the years, several different types of polymers have been studied as potential delivery tools. Polymeric NPs were initially based on poly(methyl methacrylate) (PMMA), polyacrylamide, polystyrene, and polyacrylates. Those were non-biodegradable, and therefore, they showed several issues in terms of disposal, degradation, and toxic accumulation in tissues. Thus, it was necessary to promote the elimination of NPs via feces or urine or physically remove them. Moreover, the use of these polymers was often associated with chronic toxicity and inflammation. These disadvantages led to a shift in the focus on biodegradable polymers. Biodegradable polymers (Table 1) are both based on natural materials including albumin, alginate, chitosan and gelatin, or synthetic materials such as poly(amino acids), poly(ε-caprolactone) (PLC), poly(lactide) (PLA), and poly(lactide-*co*-glycolide) copolymers (PLGA) [35,37].

The favorable biocompatibility and biodegradability are some of the features linked to their success in drug delivery. In fact, for a suitable outcome, after administration, polymers should be disaggregated into biocompatible molecules available for metabolic pathways [64].

##### Chitosan-Based Polymers

Chitosan (poly (1,4-β-d-glucopyranosamine)) is a natural polysaccharide originating from the *N*-deacetylation of chitin that shows great potential as a biomaterial for the construction of nano-sized drug carriers and gene transfer vectors [65]. In detail, chitosan possesses mucoadhesive and antimicrobial properties, and it shows good coagulation ability and immunostimulant activity. It spontaneously forms microspheres, which could be filled with various compounds such as imaging agents and drugs. Different formulations of chitosan and its composites were investigated for medical purposes due to the possibility of sterilizing it, its biocompatibility, its ability to increase the solubility of insoluble drugs, and its safe delivery to the specific site [66]. Moreover, chitosan is degraded by enzymes, such as lysozyme and chitosanase, into non-toxic and endogenous to human body oligomers; a low charge density around neutral pH reduces its cytotoxicity [29]. These systems are preferentially designed to allow their self-assembly in the presence of the drug to be incorporated [67]. Because chitosan has a robust electrostatic affinity for anionic biomacromolecules such as DNA and RNA in saline or acetic acid solution, the degradation of these biodegradable polymers can be used as a tool to release nucleic acids into the cytosol. This delivery system is advantageous due to the ability to protect DNA and RNA from degradation, and it enables the controlled release of therapeutic compounds. Compared to viral vectors, the transfection efficiency of chitosan–DNA complexes is relatively low, and it is shown to depend on numerous factors, including the structure of the polycations used, NP size and composition, and the cell type being transfected [27,68]. Nucleic-acid-based therapy is a prominent, novel, and promising area in pharmaceutical and medical sciences, and biodegradable polymeric NPs, such as chitosan, are safe and effective as carriers, by protecting the nucleic acids from degradation and cellular uptake of nuclease, without the help of transfection agents [34,35,69].

##### Poly(lactide-*co*-glycolide) and Poly(vinyl alcohol) Polymers

Synthetic polymers have considerable potential in the development of nanocarriers because of their chemical versatility, high purity, and controlled production process as compared to natural materials. Among many biodegradable and biocompatible synthetic polymers, PLGA is successfully used and investigated due to its extensive properties. It is approved by the US FDA and the European Medicines Agency (EMA) as a promising drug delivery system for therapeutic agents such as chemotherapeutics, antibiotics, anti-inflammatory or antioxidant drugs, and proteins [70,71]. PLGA is a copolymer consisting of two different monomer units, poly(glycolic acid) (PGA) and poly(lactic acid) (PLA), which are linked together by ester linkages; the result is a linear, amorphous aliphatic polyester product. Its success is mainly related to its continued drug release compared to conventional devices. In vivo, the polymer undergoes degradation by hydrolysis with the consequent formation of the original monomers (i.e., lactic acid and glycolic acid), which are endogenous molecules also produced in normal physiological conditions that are easily processed in metabolic pathways, such as the Krebs cycle; they are removed as carbon dioxide and water, causing minimal systemic toxicity. The negative charge of the PLGA is also crucial in its activity [72] because it strongly influences the interaction between NPs and cells. Cationic surfaces promote cellular binding and uptake due to negatively charged phospholipid groups, proteins, and glycans found on surface cells. On the other hand, positively charged NPs show rapid clearance and phagocytic uptake. In contrast, anionic NPs, as well as those with a neutral surface, show a higher circulating half-life [36]. Another synthetic and biocompatible polymer extensively studied is poly(vinyl alcohol) (PVA). It is used both in clinical and non-clinical research in the industrial and medical sectors. This versatility arises from its low toxicity for human tissues and its physicochemical properties (i.e., film-forming, emulsifying, flexibility, thermostability, and water solubility) [73]. Focusing on the nanomedicine field, PVA is frequently used as an emulsifier in the formulation of PLGA NPs, due to its ability to form an interconnected structure with the PLGA, helping to achieve NPs that are relatively uniform and small [74].

Starting from these considerations, NPs are supposed to be unique tools in nanomedicine because of their several possible applications. However, to design a drug carrier it is essential to keep in mind that in the human body, there are various obstacles to overtake, and therefore it is also relevant to consider how NPs are biodistributed.

### 2.2. Biological Limits to Nanodevice Delivery

Most of the anti-tumor agents currently administered by validated therapeutic protocols are systemically distributed without preferential localization to cancer tissue. Drugs had to be administered in a high ratio to reach their target at a sufficient concentration to develop the desired effects; however, their widespread biodistribution results in both loss of anticancer effects and the development of off-target adverse effects. The drug toxicity includes immune hypersensitivity, off-target toxicity, and bioactivation or covalent modifications because many drugs are converted to reactive products, which induce immune responses. In addition, idiosyncratic responses are rare but one of the most problematic issues. Therefore, every time a new therapeutic approach is proposed, research is carried out in the most efficient way to maximize efficacy, while reducing side effects. Toxicity has been evaluated to be responsible for the retreat of ~1/3 of drug candidates, and it is a major contributor to the high cost of drug development, particularly when not recognized, until late in the clinical trials or post-marketing [75]. The possibility that any molecule has sufficient therapeutic efficacy and target recognition specificity, as well as all the tools required to bypass multiple biological barriers, is probably unrealistic. A different approach is to decouple the problem. For example, it is possible to propose drugs for their therapeutic action only and deliver them to a specific microenvironment by vectors that can be preferentially concentrated at desired body locations through the concurrent action of multiple targeting mechanisms; this approach increases drug efficacy and safety, including the use of NPs as drug delivery systems for cancer therapeutic approaches [25,75]. Since Paul Ehrlich, considered the “father of chemotherapy”, suggested the concept of a “magic bullet”, i.e., “a drug that selectively attaches to diseased cells but is not toxic to healthy cells” approximately a century ago, a great deal of interest has been channeled to overtake several obstacles that drugs meet before reaching their target [76].

The kidneys represent an obstacle to the efficacy of NPs because of the risk of premature elimination, in which NPs are eliminated prior to arriving at the target tumor tissue. In fact, they are responsible for filtering circulating blood, and therefore the barriers involved in kidney filtration need to be considered when designing NPs. After passing through the fenestrated endothelium, NPs must pass through the glomerular basement membrane. Size, charge, and shape are all characteristics that affect the clearance of NPs in kidneys. Spherical NPs with diameters less than 6 nm were shown to have greater renal clearance than those with diameters greater than 8 nm. The glomerular basement membrane is negatively charged, and therefore cationic NPs of 6–8 nm exhibit greater clearance than those negatively charged or neutral of the same size [70,77,78]. Moreover, highly cationic NPs are rapidly cleared from circulation by kidneys to a greater extent than highly anionic NPs. In contrast, neutral NPs, as well as those with a slight negative charge, show significantly prolonged circulating half-lives. This translates to improved accumulation in tumors, which in turn has led to recent research efforts aimed at functionalizing NPs with zwitterionic surfaces.

As mentioned before, an important feature that affects in vivo nanocarriers’ fate is the size; in fact, larger particles (>200 nm) are rapidly engulfed by phagocytic cells found in the liver and spleen [79,80]. Injected NPs are not “self”, and the body tends to eliminate these foreign bodies. The most efficient way to recognize and bind these “non-self” agents is the attachment of serum proteins, particularly opsonins, creating a “biomolecular corona” around the NPs [81]. At this point, as exampled in Figure 4, NPs coated by opsonins are recognized by the Mononuclear Phagocyte System (MPS), which is located in the liver, the spleen, the lungs, and inflammatory tissue [78].

This process causes a release of cytokines, increasing NPs’ clearance from the bloodstream and local inflammation of the tissue. In this way, it is demonstrated that NPs are rapidly eliminated, so they cannot exercise their therapeutic action. Starting from these considerations, it is obvious to focus on defining methods to make NPs stealthy and limit the process. Surface modifications of NPs may permit escape from the MPS and prolong their circulation time in the bloodstream while preventing damage to normal tissue [78].

Successful strategies to reduce and avoid the engulfment by the MPS were developed, including the NPs’ surface functionalization with sterically shielding and hydrophilic polymers, in particular PEG. This strategy prevents, minimizes, or modifies the protein absorption, resulting in less vulnerability and prolonged circulation half-life [82,83,84]. In addition to PEG, alternative NPs’ surface modification approaches are under investigation, including those involving endogenous components, such as proteins and lipids.

#### 2.2.1. Protein Corona

NPs, when injected into the bloodstream, interact with more than 3000 circulating proteins [85]. Due to the high surface-area-to-volume ratio, NPs attract plasma proteins (e.g., albumin, complement proteins, fibrinogen, and immunoglobulins) onto their surface to form a coating layer called protein corona (PC). Its formation is a time-dependent process that affects the biological identity of NPs and, consequently, their functionality and safety [86]. The process is based on the spontaneous absorption of circulating protein around the NPs, immediately after their administration in a biological environment. Interestingly, the protein corona is not a solid and fixed layer but its composition changes over time depending on the plasma proteins’ concentration. However, multiple factors control the composition of the PC, and in general, according to the binding affinity and rate of exchange of proteins from the NPs’ surface, it can be divided into two parts: a “hard” corona and “soft” corona (Figure 5).

In this regard, binding and exchanging are the pillars of this dynamic process that can be described by the “Vroman Effect”. According to this phenomenon, blood proteins with smaller sizes (e.g., albumin) initially interact first with NPs, forming a coating called “soft” corona. They are loosely attached to the NPs’ surface and rapidly relocate from the NPs’ surface. This promotes the replacement of less abundant proteins with those with higher affinity. The “hard corona” is defined by proteins that have been proven to have a high binding affinity with the surfaces of NPs and are generally located in the inner layer of the PC. “Soft” PC constitutes the outer layer of the PC. It is fickle, since the existing proteins can be easily exchanged with others in the biological environment, depending on the abundance of proteins in biological fluids and their direct contact. Furthermore, a “soft” PC may interact with the formed “hard” PC layer on the NPs’ surface via weak protein–protein interaction. The “hard” PCs’ protein exchange rate from the NPs’ surface is slow, but this process plays an essential role in defining the NPs’ activity and biological fate [87,88,89,90]. This process of adsorption of proteins on the surfaces of NPs is dynamic and it is associated with the continuous adsorption/desorption equilibrium of the proteins on and off the NP surfaces [89].

The extent of the coating can also change according to NPs’ physicochemical properties. For example, anionic NPs are less susceptible to this phenomenon, unlike the cationic ones, in which the positive charge promotes the absorption of opsonins [91]. In addition to the nanomaterials, a focal role is certainly performed by the nature of the absorbed proteins, which can be classified as opsonins and dysopsonins. Opsonins (i.e., Ig, coagulation, and complement proteins) promote recognition by MPS. In contrast, dysopsonins, such as apolipoproteins and albumin, reduce opsonins’ absorption, conferring stealth and decreasing clearance [80]. In line with this, a promising surface modifier should be albumin. Generally, human serum albumin (HSA) and bovine serum albumin (BSA) are two types of serum proteins used in constructing or coating NPs, due to properties such as high solubility and long half-life in the blood. Albumin is the most abundant plasma protein in the human body, involved in the transport of nutrients and other proteins through the bloodstream. Since albumin is an endogenous protein, it has been widely used as an excipient in clinical formulations mostly for its biocompatibility and less immunogenicity [92]. Several features contribute to using albumin in drug delivery. For example, albumin is exploited by many cancer cells as a source of energy and nutrients, due to the enhanced uptake via macropinocytosis, which is an endocytic process. Its role as dysopsonin in the protein corona contributes to reducing the binding and absorption of other plasma proteins and, consequently, the MPS uptake [93]. For this reason, a preformed protective albumin corona should limit these non-specific interactions and decrease the complement activation. In addition to reducing clearance, albumin also has structural advantages; both amino and carboxylic groups can be employed to functionalize NPs’ surface by targeting ligands to facilitate transport and accumulation to the tumor site [89]. Therefore, beyond the improvement of the NPs’ half-life, it is crucial to consider the NPs’ distribution inside the tumor tissue.

#### 2.2.2. The Impact of Targeting

Despite the advancements in cancer therapy, chemotherapeutic-related toxicity still remains an obstacle. The cause is the inability of these agents to target the neoplastic areas without impact on healthy tissues. Consequently, they spread out after administration, resulting in off-target effects and systemic toxicity. Starting from these considerations, the research has focused on developing targeting systems capable of overcoming these pitfalls [25]. Targeting strategies should be both passive and active, sometimes associated with stimuli-sensitive mechanisms. In this way, the drug release occurs upon internal (i.e., patho-physiological/patho-chemical conditions) or external (i.e., physical stimuli such as temperature, light, ultrasound, and magnetic force) triggers. The result is enhanced intracellular delivery and consequently suitable therapeutic outcomes by using conditions present in the neoplastic tissue, minimizing systemic exposure, and limiting potential harm [94].

##### Passive Targeting in Tumor Microenvironment

Solid tumors are characterized by a heterogeneous vasculature for size and distribution, constituted by a central avascular/necrotic region and a vascularized peripheral region, with discontinuous endothelium in the micro-vessels. In fact, depending on the anatomic region of a tumor, the pore size of the endothelial junctions varies from 100 to 780 nm with a mean of approximately 400 nm (while normal vasculature is characterized by pores smaller than 10 nm), characterizing the leaky microvasculature of tumors; it is also characterized by a disrupted basement membrane, abnormal branching, and enlarged inter-endothelial gaps, with an associated breakdown of tight junctions between endothelial cells. Moreover, tumor vasculature lacks lymphatic drainage and is rich in fenestrations and poor in pericyte coverage. All these features contribute to the enhanced permeability and retention (EPR) effect, firstly described by Maeda and co-workers [95]; the tumor vasculature architecture allows extravasation and selective accumulation of nanodrugs in the tumor interstice via a passive targeting mechanism [25,96,97]. Thanks to their size, small molecules diffuse freely in and out of tumor blood vessels and thus do not accumulate in tumors as much as macromolecules do over time. These large gaps between endothelial cells facilitate the extravasation of particulate material from the surrounding vessels into a tumor (Figure 6), but EPR-dependent drug delivery is always compromised by regional blood flow rates, molecular size, polarity, and complexation to serum proteins, which are all factors that reduce the capability of bypassing the membranes of the endothelium.

However, some stimulators result in enhanced vascular permeability and extravasation of macromolecules and thus increase the EPR effect. EPR augmenting factors include vasoconstrictors to raise systemic blood pressure, free radicals that affect the integrity of vascular endothelium, and vascular permeability promoters [98]. Therefore, it is clear that the EPR effect is a very heterogeneous phenomenon, varying dramatically from tumor to tumor and from patient to patient. As a consequence, passive targeting strategies have shown several limitations; an alternative is represented by the investigation of tumor-selective active targeting nano-formulations that can maximize the accumulation at sites of interest [99,100].

##### Active Targeting in the Tumor Microenvironment

Active targeting systems are developed for selectively delivering cargo to cancerous cells without harming normal cells. The molecular recognition of the over-expressed receptor/antigen on cancerous cells is exploited by targeting molecules on the surface of nano-systems, and to benefit from this strategy, it is imperative that the tumor-associated antigen is present on the targeted cells and accessible to bind the NPs. It is also important that tumor antigen localization and expression remain adequate in the different cancer cell populations and throughout the treatment. NPs, and in particular polymeric NPs, allow for versatile modification possibilities for the assembly of well-defined multifunctional structures that can act as functional platforms. In fact, slight variations in polymeric composition, as well as ligand surface functionalization, can facilitate the targeting ability of NPs in biological systems. Active drug targeting involves the use of a variety of ligands and, parallelly, of antigens differentially over-expressed, able to direct NPs to many biological targets, primarily represented by both in the tumor cells, vessel endothelium in the tumor microenvironment, or in other diseased tissue (Figure 7) [25,96,101,102].

Moreover, margination dynamics to endothelial walls is a crucial NP design consideration. Indeed, the small spherical particles are found in a particular region of the vessel known as the cell-free layer, which results from the tendency of red blood cells to accumulate preferentially within the core of a vessel. The proximity of NPs to vessel walls favors particle–cell binding and receptor–ligand interactions in active targeting strategies and enables extravasation through the fenestrated vasculature of tumors. The size and geometry of the construct highly influence the dynamics in blood vessels [16,103].

The main purpose of the targeting ligand is to enhance the uptake of NPs into target cells to improve the therapeutic efficacy as compared with non-targeted NPs. This suggests that, while the biodistribution would be strictly related to properties linked to the structure of the NPs, the targeting ligand is essential for enhancing both cell recognition and cell uptake at target sites [25,70]. As a result, an important step in the design of the nanocarrier is represented by choice of an appropriate targeting ligand.

Different classes of targeting agents can deliver NPs in specific tissues, including proteins (e.g., antibodies or their fragments), aptamers, or small molecules such as vitamins and peptides. Antibodies (Ab) are the most common molecules used as therapeutic and targeting ligands due to their high specificity and affinity. Indeed, their use showed a new perspective on disease treatment [76]. Limits to be considered in their application are related to their size and potential immunogenicity. These restrict their density on the NPs’ surface and increase the diameter of NPs. However, the improvement of molecular technology allows circumventing these inconveniences through engineered antibodies, including single-chain variable fragments (scFv) or antigen-binding fragments (Fab) [36]. However, it is important to keep in mind that, depending on the type of ligand–receptor interaction, the type and the degree of cellular internalization would be different; moreover, proteins in biological serum were reported to shield targeted NPs by the formation of protein corona, which may impact the targeted delivery [104,105].

## 3. Therapeutic Approaches to B-Cell Malignancies

B-cells are a subtype of white blood cells that play an essential role in the immune system. In malignancies, the physiological turnover of these cells fails, and they grow at an abnormal rate when the body does not need them. Since B-cell malignancies develop from different stages of development, these constitute a heterogeneous group of pathologies [106]. For this reason, they possess intra- and inter-patient differences that can significantly influence both the selection and the duration of the treatment [107].

In recent years, many different therapeutic approaches have been investigated, such as Chimeric Antigen Receptor (CAR) and immune checkpoint inhibitors (ICIs), which have been shown to improve outcomes in patients with refractory B-cell tumors [108]. Alternatively, strategies that use innate immune cells to target B-cell malignancies represent an attractive and rapidly evolving field. A variety of options exist for mAb therapy, including various Fc engineering strategies to enhance antibody-dependent cellular cytotoxicity (ADCC), antibody-dependent cellular phagocytosis (ADCP), and complement-dependent cytotoxicity (CDC) [109]. However, remarkable clinical responses have been achieved in specific subsets of B-cell malignancies through the treatment with CAR-T cells, even if there are, unfortunately, many limitations to therapeutic efficacy, including severe life-threatening toxicities, modest anti-tumor activity, antigen escape, and limited trafficking. Moreover, the function of CAR-T cells is critically affected by the interaction of the host and tumor microenvironment with CAR-T cells [110]. On the other hand, mAb therapy is, by definition, precise and selective for the selected antigen but is usually used in combination with dose-intensive chemotherapy regimens that are not cell-specific, leading to the development of off-target side effects [111].

Over the years, research has focused on the identification of “next-generation” approaches benefiting from both chemotherapy regimens and immunotherapy. Results in this research field provide a rationale for investigating further targeting antigens in the treatment of B-cell disorders, remembering that the ideal one would be a broad-spectrum antigen in order to expand the target pathologies.

Since the purpose is to guarantee the therapeutic effect given by the chemotherapeutic payload and simultaneously avoid side effects, “next-generation” approaches can be represented by nanodevices equipped with an active targeting mechanism; recent studies on this kind of nanomedicine proved to be helpful in affecting only neoplastic cells, preserving the viability of the healthy ones [112].

On this basis, it is interesting to develop a strategy that takes advantage of “next-generation” treatments, which combines the knowledge about NPs with drug efficacy. This alternative approach, based on nanocarriers, offers the possibility to take advantage of the specificity and selectivity of the targeting mechanism, meanwhile encapsulating the drug or the therapeutic molecule, avoiding off-target side effects, and reducing the costs of disease management because the doses required to achieve the same efficacy as untargeted drugs might be lower [14]. To this end, different classes of therapeutic molecules can be investigated, alone or in combination.

### 3.1. Tumor B-Cell-Associated Antigens

Each B-cell malignancy is associated with a specific developmental stage and specific surface antigens’ expression. Surface molecules have particular importance in identifying the degree of maturation for diagnosis, prognosis, and the best treatment options; besides, they can be used as targets for specific clinical interventions [113,114]. Ideal tumor antigens should be abundant, accessible, and homogeneously, consistently, and exclusively expressed on the surface of target cells; moreover, antigen secretion should be minimal. All these features guarantee the activity of the targeting agent because of the antibody’s specificity and the minimal side effects. Indeed, Ab-based immunotherapy is a type of cancer treatment that helps the host immune system fight cancer, thanks to the fact that the modified immunotherapeutic antibodies bind to the tumor antigens, marking and identifying the cancer cells as non-self structures for the immune system [115].

Concerning B-cell malignancies, surface antigens such as CD19, CD20, CD22, CD30, CD38, and CD52 are currently the primary targets for immunotherapy. In detail, CD19 and CD20 are markers specific for cancer B-cells, while CD30, CD38, and CD52 are generally expressed on the surface of mature B and T lymphocytes but also on malignant lymphocytes. CD22 is a regulatory component of the B-cell Receptor (BCR) complex, expressed exclusively in pre-B- and mature B-cells. Since CD22 endocytosis can be triggered efficiently, antibodies and antibody-based immunotoxins against CD22 have raised great interest in the treatment of some subtypes of B-cell malignancies [116]. Other tumor-associated antigens under investigation are represented by CD30 and CD52. The first one is a specific marker of Hodgkin lymphoma and anaplastic large cell lymphoma, but it is expressed also in B-cell lymphomas, including diffuse large B-cell lymphoma (DLBCL) and primary mediastinal large B-cell lymphoma [116]. On the other hand, the presence of CD52 has been demonstrated on many cells, such as B and T lymphocytes, where it is strongly expressed, as well as on monocytes and macrophages (which also express low levels of CD52) and on most malignant lymphoid cells [117]. Nevertheless, historically, the attention was focused on other most promising antigens.

#### 3.1.1. CD20 Antigen

Among B-cell-related antigens, CD20 was the first molecule against whom mAbs were developed [30,118,119,120]. The human CD20 gene is encoded by eight exons located on chromosome 11, and the mouse gene is found in an evolutionarily conserved region of chromosome 19 (showing 73% homology). The CD20 antigen (Figure 8) is absent in early stem cells or in plasma cells but is present on mature B-cells through plasma, making this antigen fundamental for B-cell ablative therapy [121].

This therapy allows for the elimination of all cancer cells and, at the same time, saves early stem cells. In this way, it is possible to restore the B-cell population after the treatment. The first mAb to receive approval by the FDA for relapsed/refractory NHL in 1997 was rituximab (Rituxan^®^), an anti-CD20 chimeric mAb, and since its approval, rituximab has been employed for use the treatment of numerous other B-cell malignancies, as well as autoimmune conditions, including rheumatoid arthritis [121]. Therefore, for nearly 20 years, rituximab-based therapies have demonstrated their dominance in this field; on the other hand, if a patient experiences relapsed or refractory disease after rituximab-based treatment, there are limited options for salvage [116]. For this reason, other antigens are under investigation, and nevertheless, the focus of immunotherapy development for over 30 years was set on the most characterizing antigen of B-cells, that is, CD19.

#### 3.1.2. CD19 Antigen

CD19 is a 95 kDa member of the Ig superfamily and is expressed nearly exclusively on B lymphocytes, together with other cell-surface regulatory molecules. CD19 is expressed by almost all B-cell developmental stages, and it is a critical co-receptor for B-cell signaling, which acts in signaling, cell activity, and proliferation [122]. In particular, CD19 is a cell-surface glycoprotein, and it exists in a complex with CD21 and CD81 and provides a link between innate and adaptive immunity, due to its ability, with the help of CD21, to bind the complement C3 cleavage product C3d enabling CD19 and the BCR to interact. These interactions permit the reduction in the number of antigen receptors that need to be stimulated to activate the cell, reducing the threshold required for B-cell proliferation in response to a given antigen [123]. Since CD19 plays a pivotal role in BCR signaling and its overall expression pattern from pro-B-cells until the terminal differentiation to plasma cells (Figure 8), it is not surprising that abnormal CD19 expression on B-cells is associated with autoimmune diseases and development of malignancies [122,124]. For these reasons, CD19 represents a more exploitable antigen than CD20.

Pharmaceutical companies are actively pursuing anti-CD19 strategies; indeed, anti-CD20 therapies do not directly target early B-cell differentiation stages (i.e., ALL) or B-cell malignancies resistant to anti-CD20 approaches. Although CD19 has been a focus of immunotherapy development for many years, it is only recently that durable therapeutic responses have been achieved with CD19-directed approaches. Even with some limitations, several anti-CD19 therapies are currently in clinical trials, including mAb, antibody-targeted cytotoxic drug conjugates (ADC), bispecific antibodies, and CAR-T cells [124,125]. For instance, given the substantial improvement in overall survival by the mAb anti-CD20 rituximab in ALL patients with significant CD20 expression (≥20%), the implementation of CD19 mAb, potentially reaching a broader spectrum of patients, is a reasonable strategy [123]. Although therapeutic antibodies ensure the specific effect and improve survival, compared to chemotherapy alone, an assessment of the disadvantages is necessary due to the frequency of incomplete response and resistance phenomena [126].

## 4. Clinical Application of Nanomaterials to B-Cell Malignancies

In contrast to the traditional therapy of B-cell malignancies based on dose-intensive chemotherapy regimens, it is interesting to develop a strategy combining drug knowledge with new nanocarriers. This alternative approach is expected to be more specific, thanks to the specific targeting mechanism, and at the same time with fewer side effects. Therefore, it is possible to selectively treat the pathology by encapsulated chemotherapy or interfere with the dysregulated process, such as the over-expression of miRNA, reducing its level and stopping the proliferation of cancer cells. A novel therapeutic approach based on biodegradable NPs equipped with a specific targeting antigen (e.g., anti-CD19 or anti-CD20) and containing the active agent can specifically identify malignant B-cells and obtain a more specific and safer treatment. In addition, NP formulations can be used to enhance the efficacy of immunotherapy in cancer treatments by reversing an immunosuppressive state that may be due to multiple factors, including the presence of myeloid-derived suppressor cells, macrophages, and regulatory T-cells.

From a clinical perspective, several NP-based drugs have already received FDA approval for the treatment of hematologic malignancies (Figure 9). The first FDA-approved liposomal nanocarrier was Doxil^®^, in 1995, which is a PEGylated liposomal formulation loaded with doxorubicin. The coating of PEG protects the liposome from detection by phagocytic cells, allowing a prolonged circulation time. Moreover, compared to the free drug, the liposomal formulation is demonstrated to minimize potential side effects of the drug due to the protection given by the structure of the carrier [27]. In the following years, other liposomes were approved by the FDA: Caelyx (1996, for the treatment of metastatic breast cancer, advanced ovarian cancer, progressive multiple myeloma, and AIDS-related Kaposi’s sarcoma), vincristine sulfate liposomal (2008, for the treatment of ALL and other lymphoid malignancies), Vyxeos (2018, for the treatment of adults with newly diagnosed, therapy-related acute myeloid leukemia (AML) or AML with myelodysplasia-related changes), and, more recently, Zolsketil (2022, for the treatment of metastatic breast cancer, advanced ovarian cancer, progressive multiple myeloma, and AIDS-related Kaposi’s sarcoma). Only one of them is based on vincristine, while the others convey anthracyclines. The first one is a well-known alkaloid employed in the treatments of ALL and other lymphoid malignancies but with limited clinical use due to unpredictable pharmacologic characteristics; parallelly, treatments with anthracyclines are associated with an augmented incidence of cardiotoxicity. On these bases, nanocarriers result to be fundamental to encapsulating and protecting the loaded drug, maintaining the efficacy of the treatment, and avoiding side effects [127,128,129]. In addition, about 70 clinical trials are currently under investigation: 9 of them are completed, 32 are under recruitment, and others are approved but in different phases of development.

## 5. Conclusions and Future Perspectives

B-cell malignancies possess intra- and inter-patient differences that can significantly influence both the choice and duration of treatment [107]. In recent years, the focus was set on “next generation” treatments, capable of overcoming the typical limits of standard treatments, which have been effective but with a lot of limitations. Indeed, chemotherapy is known to be debilitating, and its lack of specificity leads to significant side effects.

Therefore, nano-systems able to contain and deliver drugs primarily to the desired site have become increasingly popular, and several materials, structures, and approaches are still under investigation. However, in general, the use of NP-based drugs in routine clinical practice still requires specific regulatory guidelines, detailed chemical and physical characterization, and in-depth investigation of potential toxicity effects. In addition, many NP-based drug approaches remain stuck at the preclinical level without the opportunity to be evaluated in a clinical trial. This is often due to the difficulty of incorporating them into already established chemo-immunotherapy combinations. In addition, the efficacy of nanodrugs for targeted treatment approaches depends on the expression level of the chosen antigen, and the targeting ability of nanodrugs may be reduced by the formation of PC in the physiological fluids. Further efforts need to be made to efficiently introduce drug approaches based on NPs into routine clinical practice in the treatment of hematological diseases.

## Figures and Tables

**Figure 1 pharmaceutics-14-01965-f001:**
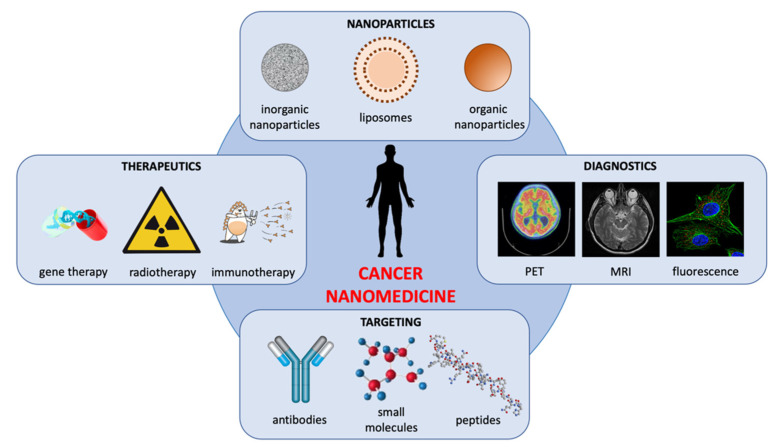
Nanodevices in imaging and therapies. The compatibility between nanoparticle size and the biological systems, coupled with the ability to tailor their physicochemical properties, enhances a personalized approach to disease management, and therapies could considerably improve the diagnostics and therapeutics of various cancers.

**Figure 2 pharmaceutics-14-01965-f002:**
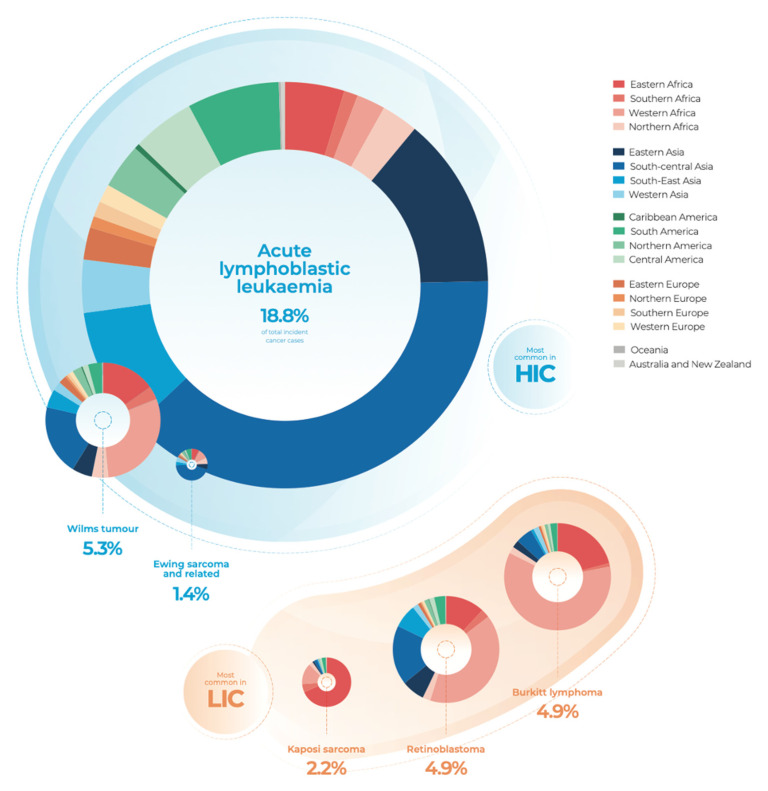
Estimated total incident cancer cases in 2015 by the top 15 specified diagnoses and regions. HIC: high-income countries; LIC: low-income countries (adapted from World Health Organization, 2021).

**Figure 3 pharmaceutics-14-01965-f003:**
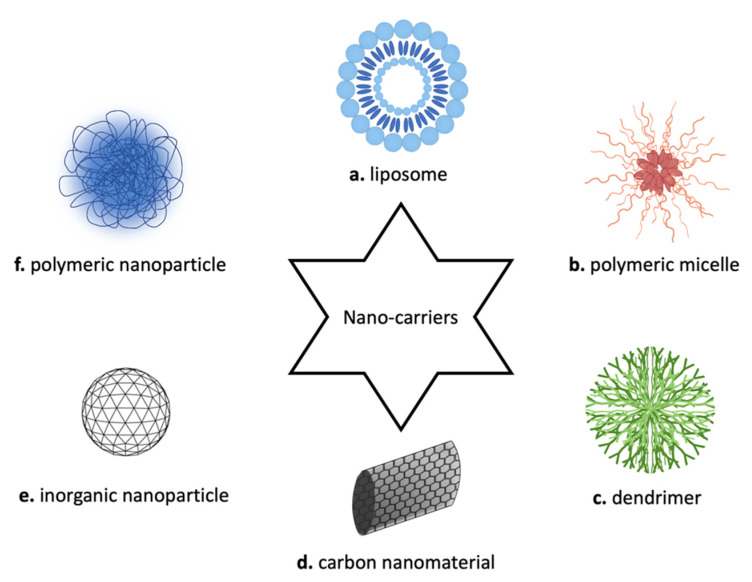
Schematic representation of nanocarriers for cancer therapy. In commerce, there are different types of nano-vectors, such as (**a**) liposome, (**b**) polymeric micelle, (**c**) dendrimer, (**d**) carbon nanomaterial, (**e**) inorganic nanoparticle, and (**f**) polymeric nanoparticle.

**Figure 4 pharmaceutics-14-01965-f004:**
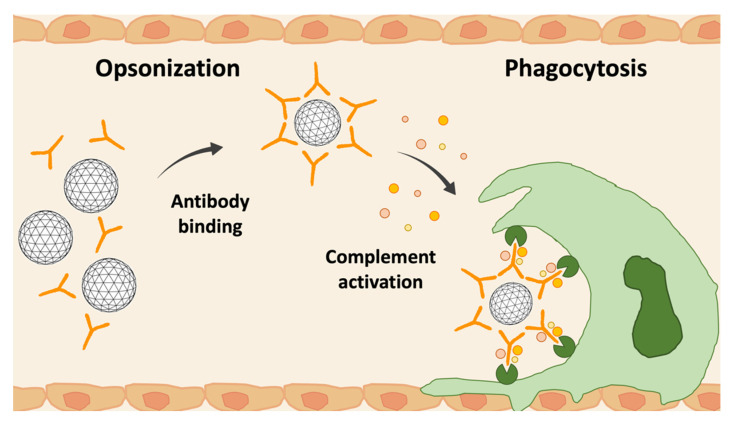
Opsonization process. Serum proteins are absorbed on the NPs’ surface leading to the opsonization process; In liver vessels (endothelial cells in orange), opsonized particles are subsequently recognized through receptors on phagocytic cells (green cell) and then internalized.

**Figure 5 pharmaceutics-14-01965-f005:**
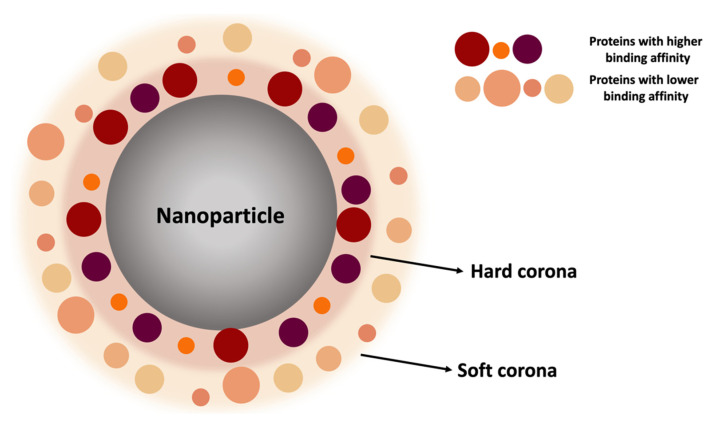
The duality of the protein corona. Proteins with a smaller size and lower affinity constitute the so-called “soft” corona. They are loosely attached to the NPs’ surface and relocate from the NPs’ surface rapidly. This promotes the replacement of less abundant proteins with those with higher affinity belonging to the “hard” corona.

**Figure 6 pharmaceutics-14-01965-f006:**
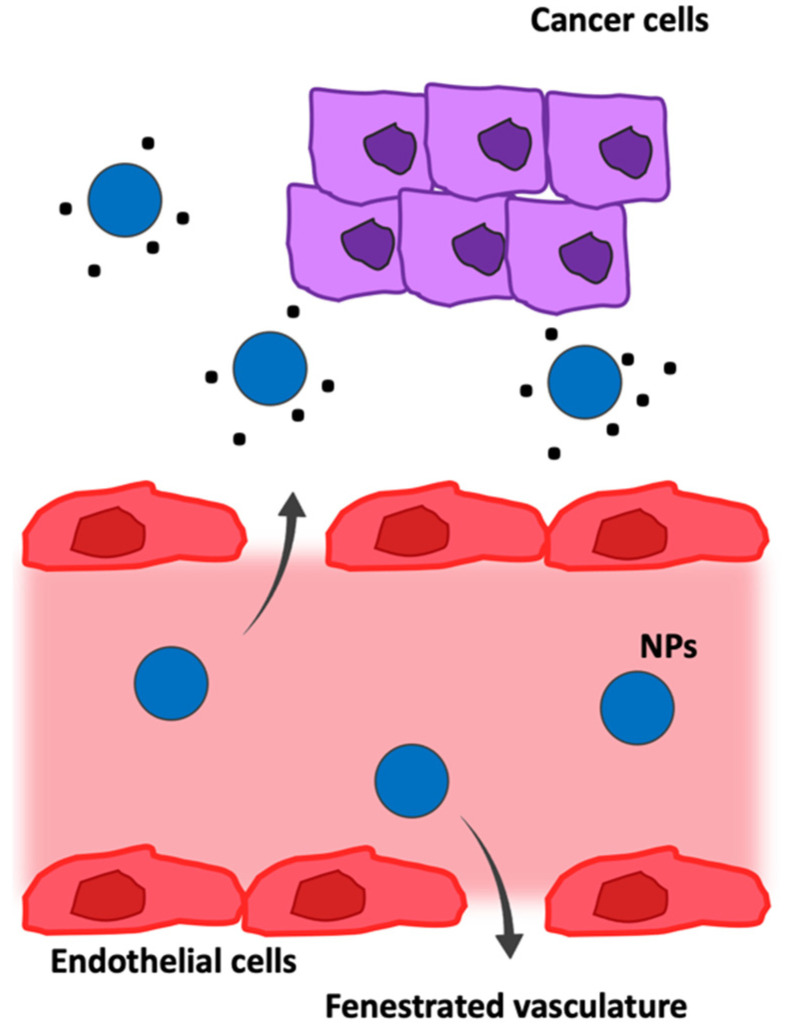
Passive targeting. In passive targeting via bloodstream (pink), nanoparticles (blue) are designed for transport through leaky vessels (red) and the unique intra-organ pressures of tumors (purple).

**Figure 7 pharmaceutics-14-01965-f007:**
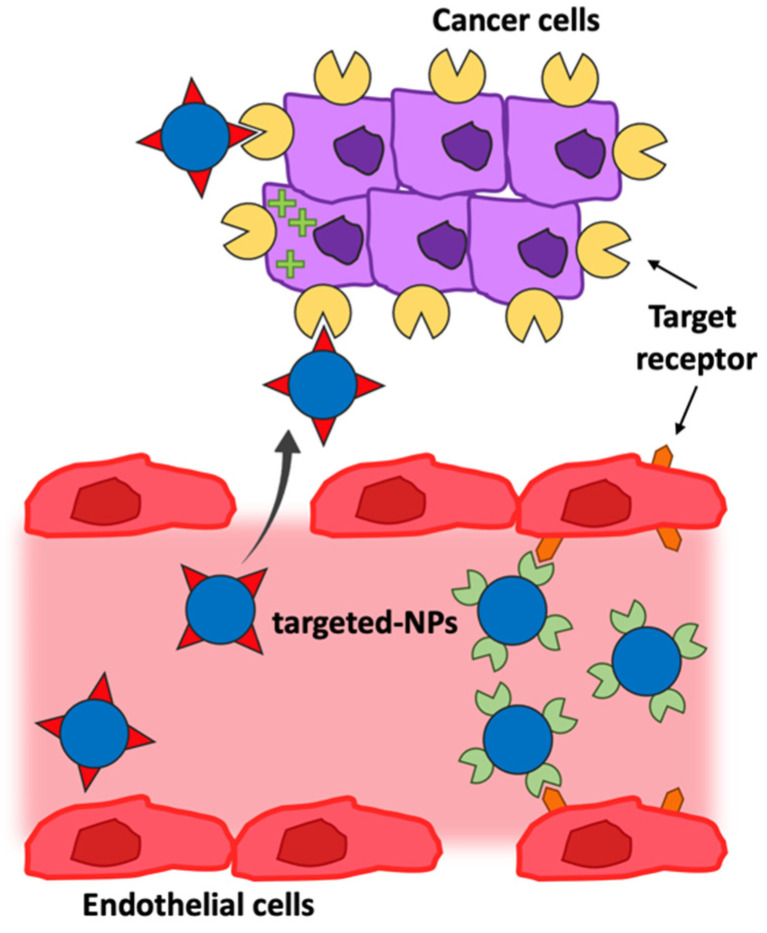
Active targeting. In the active targeting of cancer cells (tumor-associated antigens in yellow) or tumor-endothelium (antigens in orange), nanoparticles (blue) are designed to bind specific biological structures through the molecular recognition (targeting in red or green) of surface-bound ligands.

**Figure 8 pharmaceutics-14-01965-f008:**
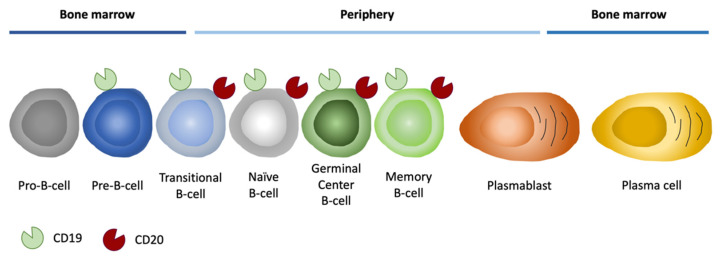
Expression of CD19 and CD20 in B-cell lineage. Illustrative representation of B-cell differentiation and maturation. CD19 antigen is expressed from pro-B-cells to plasma cells, hence CD20 from pre-B-cells to plasma cells.

**Figure 9 pharmaceutics-14-01965-f009:**
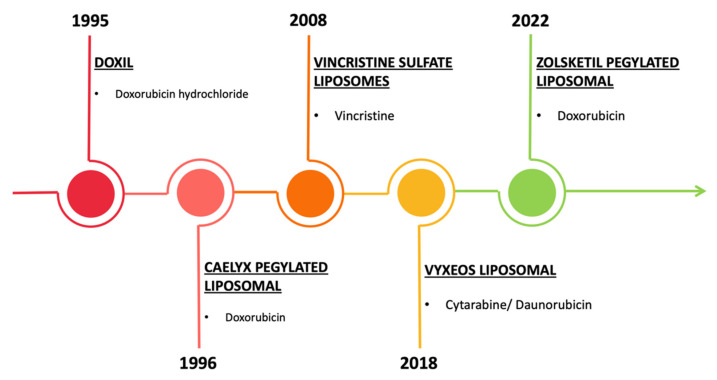
EMA’s approved NP-based drugs. Illustrative representation of the NP-based drugs approved by the EMA to date.

**Table 1 pharmaceutics-14-01965-t001:** Summary of natural and synthetic polymers and their advantages.

Biodegradable Polymers		Advantages	Ref.
**Natural polymers**	Chitosan	Abundant in nature Biodegradable Inexpensive Mucoadhesive Non-toxic	[38,39,40,41,42]
	Alginate	Able to form gel Biodegradable Low immunogenicity Mucoadhesive Non-toxic Polyanionic	[43,44,45,46]
	Hyaluronic Acid	Biocompatible Biodegradable Low immunogenicity	[47,48]
	Dextran	Biodegradable Biocompatible Hydrophilic	[49,50,51]
	Gelatin	Biodegradable Crosslinking potential Hydrophilic Non-toxic	[52,53]
Synthetic polymers	Poly(Lactide-*co*-Glycolide) Acid (PLGA)	Biodegradable Controlled release kinetics	[54,55]
	Poly-ε-Caprolactone (PCL)	Biodegradable Mucoadhesive	[56,57]
	Polyvinyl alcohol (PVA)	Biocompatible Biodegradable Low toxicity Mechanical strength Thermal stability	[58,59]
	Polyamino Acids	Charge density Chirality Hydrophilic Hydrophobic Reversible crosslinking	[60,61,62]
	Pluronic	Amphiphilic Biodegradable Soluble in aqueous polar and non-polar solvents Thermosensitive	[63]

## Data Availability

Not applicable.

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
