# Peer review of "Nanocarriers as a Delivery Platform for Anticancer Treatment: Biological Limits and Perspectives in B-Cell Malignancies"

_pharmaceutics, 2022, doi:10.3390/pharmaceutics14091965_

Round 1
Reviewer 1 Report
It was a review paper about different types of nanomaterials that could be used as drug delivery systems and their application for B-cells malignancies. Here are some comments on this study that should be considered before publication:
1- Authors need to improve the quality of the abstract.
2- There are some grammatical mistakes in the text that should be corrected.
3- “Polymeric micelles (Figure 3b) are self-assembled core-shell” micelles are not core-shell structures.
4- “Among different nanostructure morphologies for drug delivery, polymeric NP-based drug delivery shows numerous advantages for controlling the release of biological factors. These advantages include the possibility to add a selective targeting mechanism, the controlled release, protection of deliverable agents, and extension of the circulation time in the body” the mentioned advantages are not specified to polymeric nanoparticles
5- “This section may be divided into subheadings. It should provide a concise and precise description of the experimental results, their interpretation, as well as the experimental conclusions that can be drawn.” please rewrite this paragraph, it is not good.
6- “Another important feature that affects in vivo nanocarriers’ fate is the size” size was also mentioned before in lines 365-368.
7- It is better to change the title, it is not completely related to what is mentioned in the main text.
8- Why did you just describe B-cells malignancies?
9- What you mentioned in the conclusion part is not completely related to that. Normally conclusion has no figure or reference. It is better to revise the conclusion part as "conclusion and future perspective".
10- Please add a new section about the clinical application of nanomaterials for the treatment of hematological cancers.
Reviewer 2 Report
The manuscript review entitled "Nanocarriers as a delivery platform for anticancer treatment: biological limits and possible applications in the hematological field" from Bozzer et al involved the comprehensive review analyses of different nanostructures for drug delivery focus on anticancer treatments. Moreover, biological limits were discussed regarding to the hematological implications.
The introduction is comprehensive and according to the focus of the manuscript, and it has revised bibliographical references to support the research.
Also, the manuscript is interesting, organize, easy to follow and focused on the topic that is of growing interest due to the potential applications of nanoparticles drug delivery systems for this worldwide pathology with high rate of death.
In addition, the authors discussed the disadvantages of these nanoplatforms from a critical point of view. This is remarkable and demonstrates the full knowledge and capacity for critical analysis of the subject they reported.
As a suggestion, it should be important to identify some structural components in Figure 6 and Figure 7 (highlighted in yellow, pdf attached).
Furthermore, I encourage the authors to check some mistakes (in yellow, pdf file attached) such as:
- Line 271: N-deacetylation. N should be in italics.
- Please check all the figures throughout the manuscript in order to keep the same size images, fonts, and image quality.
Additionally, would like to invite the authors to add the abbreviation list of words at the end of this manuscript.
I recommend the acceptance of this manuscript after the authors performed the suggested corrections/additions.
Round 2
Reviewer 1 Report
1- The quality of the abstract is not good. please rewrite it.
2- The last paragraph of section 3.1 should be rewritten.
3- "“Another important feature that affects in vivo nanocarriers’ fate is the size” size was also mentioned before in lines 365-368." this comment from the previous review is not solved, please check the text again.
4- Are the samples mentioned in section 5 related to the clinical application of nanoparticles for B-cells malignancy? If not, please replace them with appropriate samples.
Round 3
Reviewer 1 Report
-